# Influence of Bruton’s Tyrosine Kinase (BTK) on Epithelial–Mesenchymal Transition (EMT) Processes and Cancer Stem Cell (CSC) Enrichment in Head and Neck Squamous Cell Carcinoma (HNSCC)

**DOI:** 10.3390/ijms241713133

**Published:** 2023-08-23

**Authors:** Franziska Leichtle, Annika C. Betzler, Carlotta Eizenberger, Kristina Lesakova, Jasmin Ezić, Robert Drees, Jens Greve, Patrick J. Schuler, Simon Laban, Thomas K. Hoffmann, Nils Cordes, Marialuisa Lavitrano, Emanuela Grassilli, Cornelia Brunner

**Affiliations:** 1Department of Oto-Rhino-Laryngology, Ulm University Medical Center, 89075 Ulm, Germanyezic.jasmin@gmail.com (J.E.); patrick.schuler@uniklinik-ulm.de (P.J.S.);; 2Core Facility Immune Monitoring, Ulm University, 89081 Ulm, Germany; 3OncoRay–National Center for Radiation Research in Oncology, Faculty of Medicine, University Hospital Carl Gustav Carus, Technical University Dresden, 01307 Dresden, Germany; 4School of Medicine and Surgery, University of Milano-Bicocca, 20900 Monza, Italy

**Keywords:** BTK, HNSCC, EMT, CSC

## Abstract

Constitutively active kinases play a crucial role in carcinogenesis, and their inhibition is a common target for molecular tumor therapy. We recently discovered the expression of two oncogenic isoforms of Bruton’s Tyrosine Kinase (BTK) in head and neck squamous cell carcinoma (HNSCC), Btk-p80 and BTK-p65. However, the precise role of BTK in HNSCC remains unclear. Analyses of a tissue microarray containing benign and malignant as well as inflammatory tissue samples of the head and neck region revealed the preferential expression of BTK-p80 in malignant tissue, whereas BTK-p65 expression was confirmed in over 80% of analyzed metastatic head and neck tumor cases. Therefore, processes associated with metastasis, like cancer stem cell (CSC) enrichment and the epithelial–mesenchymal transition (EMT), which in turn depend on an appropriate cytokine milieu, were analyzed. Treatment of HNSCC-derived cell lines cultured under 3D conditions with the BTK inhibitor AVL-292 caused reduced sphere formation, which was accompanied by reduced numbers of ALDH1A1^+^ CSCs as well as biological changes associated with the EMT. Moreover, we observed reduced NF-κB expression as well as altered NF-κB dependent pro-tumorigenic and EMT-associated cytokine release of IL-6, IFNγ, and TNFα when BTK activity was dampened. Therefore, an autocrine regulation of the oncogenic BTK-dependent process in HNSCC can be suggested, with BTK inhibition expected to be an effective treatment option for HNSCC.

## 1. Introduction

Head and neck squamous cell carcinoma (HNSCC) is a heterogeneous group of cancer cells with distinct locations, different risk factors, and individual prognoses [1]. HNSCC tumors originate from mucosal cells in the oral cavity, pharynx, and hypopharynx.

HNSCC treatment is of significant socioeconomic importance as it ranks as the sixth most common cancer worldwide, with increasing cases each year [2]. Standard therapy typically involves radical local surgery, radiation, and chemotherapy, tailored to the primary tumor site and stage. However, the prognosis remains poor, and there is a high risk of recurrence after the primary therapy. 

First, the microenvironment of HNSCC is highly immunosuppressive, preventing recognition of the tumor by the immune system and causing therapy failure. Immunotherapeutic approaches, such as PD-1 interaction using pembrolizumab or EGFR blocking with cetuximab, has shown some efficacy, but there is a need for additional immunomodulatory therapy options to improve outcomes [3]. Additionally, one of the primary factors associated with therapy resistance and recurrence in HNSCC is the presence of residual or recurrent tumor cells following the initial treatment. These cells may possess inherent resistance mechanisms or acquire resistance during treatment, leading to tumor regrowth. [4,5]. Another contributing factor to therapy recurrence is the ability of HNSCC to infiltrate and spread to regional lymph nodes and distant sites [1,2]. Over recent years, it has become recognized that a special subpopulation of cancer cells is mainly responsible for tumor invasion, metastasis, and therapy resistance—the cancer stem cells (CSCs) [4,5]. Some of these CSCs undergo an epithelial–mesenchymal transition (EMT) by activation alterations in the expression of extracellular matrix proteins as well as epithelial/mesenchymal characteristic markers, which enable them to dissociate from the primary tumor and to invade and recolonize in distant tissues after a process called mesenchymal to epithelial transition (MET) [6]. These dynamic alterations of cancer cells are influenced by cell-intrinsic processes, like epigenetic or metabolic shifts, but also by factors released by the tumor microenvironment (TME) [6]. The TME is composed of several different cell types, like immune cells, mesenchymal stem cells, fibroblasts, myofibroblasts, and stromal cells [7], all of which are able to secrete pro-inflammatory proteins, driving tumor progression, including the EMT process [8]. Additionally, SCCs, including HNSCC tumor cells themselves, secrete pro-tumorigenic cytokines and growth factors involved in NF-κB signaling [9,10,11,12,13], which could eventually promote CSC plasticity and EMT processes. Thus, the tagging of CSCs and/or EMT processes is a challenging but promising strategy in fighting cancer.

Bruton’s Tyrosine Kinase (BTK) is a member of the Tec family of non-receptor tyrosine kinases and is expressed on cells of the hematopoietic system. BTK plays a crucial role in B cell development and function [14]. Defects in the BTK genomic locus can lead to a primary immunodeficiency disorder known as X-linked agammaglobulinemia (XLA) [15]. Defects in BTK function lead to an arrest in B cell development at the very early stages of B cell ontogeny, resulting in less than 1% of circulating B cells in the periphery. Consequently, no primary and secondary antibodies are produced. Therefore, affected individuals need life-long immunoglobulin substitution. BTK is a crucial kinase essential for B cell receptor (BCR) signaling, but it is also expressed in other immune cells, such as neutrophils, macrophages, dendritic cells, and mast cells [16,17,18,19]. Therefore, BTK is associated with both the innate and adaptive immune system. 

BTK was found to be constitutively active in defined B cell malignancies and, consequently, it has become a target for therapeutic intervention. Small molecule inhibitors of BTK, such as ibrutinib, have been developed and approved for the treatment of certain types of B cell lymphomas and chronic lymphocytic leukemia (CLL) [20]. 

In recent years, a previously unexpected function of BTK was revealed outside of the immune system; more specifically, in solid tumor tissues, where two novel oncogenic BTK isoforms were detected—BTK-p80 and BTK-p65 [21,22]. Elevated BTK expression has been found in the lung [23], gastric region [24], colon [22], prostate [25,26] breast [21], glioma [27,28], and ovarian cancer [29], as well as in HNSCC [30,31]. Therefore, the use of FDA-approved BTK inhibitors to treat solid cancers is an ongoing research topic and is being tested in multiple clinical trials [32].

We have recently shown that both oncogenic BTK isoforms, BTK-p80 and BTK-p65, coexist in tumor cells of the head and neck origin [31]. Abrogation of BTK activity inhibited tumor progression in terms of proliferation and vascularization in vitro and in vivo in our previous study [31]. Consequently, targeting BTK activity could be a promising therapeutic option for HNSCC patients. However, the role of individual isoforms for HNSCC tumorigenic processes is largely unknown. The present study aims to investigate the individual expression patterns of BTK-p80 and BTK-p65 to obtain deeper insights into their oncogenic function. Since this study revealed high BTK-p65 expression in metastatic HNSCC, the involvement of BTK in CSC enrichment and EMT processes was further analyzed. Our data suggest that BTK influences NF-κB expression as well as pro-inflammatory cytokine secretion by HNSCC tumor cells, which could have influence on tumor cell plasticity and heterogeneity in an autocrine manner.

## 2. Results

### 2.1. Differential Expression of BTK-p80 and BTK-p65 Isoforms in Tumors of HNSCC Patients

To characterize the expression of the BTK isoforms in primary tumor tissue, a commercially available head and neck tumor tissue microarray (TMA), containing 48 cases of inflammatory, benign and malignant tumor tissues of the neck, oronasopharynx, larynx and salivary glands, was analyzed. Immunohistochemical staining for BTK-p80 and BTK-p65 expression was performed using isoform-specific antibodies: the 7F12H4 antibody to detect the BTK-p80 isoform [31] and the BN30 antibody, specific for the BTK-p65 isoform [22,23]. Staining results revealed medium or high BTK-p80 expression in malignant HNSCC tissue (45.9%) compared with inflamed (0%) and benign tissue (5.9%) (Figure 1A). At the same time, 39.6% of malignant tissue expressed BTK-p80 at a low level. These data indicate that expression of BTK-p80 is likely a common feature (>80%) of tumors of the head and neck region. Moreover, subdivision of head and neck tumors into adenoid-cystic carcinomas (ACC), squamous cell carcinomas (SCC), and others showed that BTK-p80 expression is a characteristic feature of SCC, given the 65.4% of specimens expressing BTK at a medium or high level (Figure 1B), whereas BTK-p65 is more highly expressed in ACCs and other head and neck carcinomas. Moreover, the expression of BTK-p65 is not restricted to tumor tissues only. Our findings show that this isoform is abundantly expressed in 1/3 of other pathological processes like inflammation or in benign tissues (Figure 1C,D). Most interestingly, over 80% of analyzed metastatic head and neck tumor cases revealed medium or high BTK-p65 staining (Figure 1C,D), suggesting an oncogenic function of BTK-p65 in metastatic processes.

### 2.2. Morphological Changes in Two-Dimensional Cultivated Tumor Cells from HNSCC after Treatment with BTK Inhibitor Are Associated with EMT Processes 

Since our data suggest an involvement of BTK-p65 in the metastatic process, which is associated with CSC plasticity and the EMT, we first treated two-dimensional (2D) cultures of UDSCC1 cells with AVL-292 in order to investigate possible morphological changes under BTK inhibition. AVL-292 binds to Cys481 in the ATP-binding domain of BTK and therefore inhibits all BTK isoforms. Notably, we have previously shown that AVL-292 strongly inhibited HNSCC cell proliferation [31]. After 48 h of treatment, a reduced cell number and a more epithelial phenotype compared with untreated cells were observed microscopically. In addition, mesenchymal cell extensions were less developed (Figure 2A).

To gain more insights into the involvement of BTK on the invasive and metastatic potential of HNSCC cells, we next analyzed the effect of BTK inhibition on known EMT markers at the protein level. We observed a slight increase (although not significant) of E-Cadherin expression upon treatment with AVL-292 in UDSCC-1 cells. In parallel, the Slug expression decreased significantly in a dose-dependent manner (Figure 2B).

### 2.3. Morphological Changes in Three-Dimension Cultures of HNSCC after Treatment with BTK Inhibitor Are Associated with EMT Processes

To better simulate the complex tumor microenvironment (TME), we next tested the effects of BTK inhibition on the EMT in three-dimensional (3D) cultures of HNSCC tumor cells, employing the sphere formation assay on ultra-low attachment (ULA) surface plates. The treatment of HNSCC cell lines UDSCC1, -5, and -6 with the BTK inhibitor AVL-292 also resulted in morphological changes under 3D cultivation conditions. The application of the BTK inhibitor caused a general decrease in the formation of spheres. More precisely, the larger spheres (with diameters of 70–110 µm) were significantly reduced. Conversely, an increase in the number of smaller spheres (with diameters of 40–70 µm) was detected (Figure 3A). These observations were accompanied by changes in the spheroid morphology, which were characterized by a loosening of cell-to-cell contacts, resulting in a lower density and a “cloud-like” appearance of the observed spheroids. Overall, treatment with the BTK inhibitor AVL-292 induced significant morphological alterations, including changes in the size, number, and morphology of HNSCC tumor cells cultured under 3D conditions. Also, a significant and dose-dependent increase in the expression of E-Cadherin, accompanied by a reduction of Slug expression, could be detected (Figure 3B). Similar effects were observed in all the HNSCC cell lines analyzed (UDSCC1, -5, and -6;).

Given that similar results using 2D and 3D culture systems were observed, we assessed whether the transcriptome and proteome under these different culture conditions would remain the same. Therefore, three different HNSCC cell lines, UDSCC1, -5, and -6 were either cultured under 2D or 3D conditions, left untreated (DMSO control), or treated with the BTK inhibitor AVL-292. After harvesting, RNA was prepared and subjected to RNA sequencing. Surprisingly, clear differences between the 2D and 3D samples, both with and without BTK inhibition, were observed (Figure 4). There was no overlap (0%) within the top 50 differentially expressed genes (DEG) between 2D and 3D cultures. Comparing DEGs of the top 10 up- and downregulated molecular functions, a higher expression of cytokine activity and cytokine receptor binding was detected under 3D conditions. Interestingly, analyzing the top 10 biological processes, genes associated with epidermal cell differentiation as well as cell cycle regulation and nuclear DNA replication were found to be overexpressed in cells cultivated in 3D. Considering these results, all further experiments were performed under 3D conditions.

### 2.4. Influence of BTK Inhibition on CTC Enrichment

Since there is a close link between EMT processes and the gain of stemness characteristics, the impact of BTK inhibition on the generation of CSC was explored. CSCs possess certain characteristics, including the expression of ALDH1A1. Therefore, UDSCC1, -5, and -6 cells cultured under 3D conditions were analyzed by flow cytometry using the ALDEFLUOR assay. After treatment with AVL-292, a dose-dependent decrease in the percentage of ALDH1A1-positive cells, defined as CSCs, was observed. The greatest reduction in CSCs upon BTK inhibition was revealed in UDSCC1. These findings suggest an influence of BTK on the population of CSCs in HNSCC (Figure 5).

### 2.5. BTK Inhibition Influences the NF-κB Pathway and Cytokine Release from Tumor Cells

The EMT is a regulated process that depends on pro-tumorigenic inflammatory cytokines, including IL-6, IFNγ, and TNFα. These cytokines are produced by stromal and immune cells but also by cancer cells themselves [33,34]. Among others, these cytokines are regulated by the key transcription factor NF-κB [9,10,11,12,13]. Besides its role in pro-inflammatory cytokine gene transcription, TNFα-induced NF-κB activity is also involved in the transcription and stabilization of EMT-related genes, like Slug [35]. 

In order to investigate whether BTK influences the EMT by interfering with the pro-tumorigenic cytokine release from tumor cells, UDSCC1, -5, and -6 cells were cultured under 3D conditions and either left untreated or treated with AVL-292 for 72 h. Quantification of total NF-κB p65 protein by Western blotting revealed a clear reduction in NF-κB expression upon treatment with AVL-292 in all the tested cell lines (Figure 6A). In line with these findings, we observed a clear reduction in known NF-κB targets and the proinflammatory cytokines IL-6, IFNγ, and TNFα involved in the EMT and CSC enrichment processes, as measured in the cell culture supernatants by ELISA (Figure 6B).

Together, these data indicate that BTK influences NF-κB activity as well as NF-κB-dependent cytokine secretion in HNSCC, which may modulate EMT processes as well as CSC enrichment in an autoregulatory manner, independent of other cytokine-producing cells present in the TME. 

## 3. Discussion

Previously, we described the co-expression of BTK-p80 and BTK-p65 in HNSCC-derived cell lines. In order to obtain additional information about their possible different functions in HNSCC tumorigenesis, the expression patterns of these isoforms were investigated by analyzing a tissue microarray (TMA). Interestingly, whereas BTK-p80 is strongly expressed in malignant tissue of the head and neck region, particularly in HNSCC, the BTK-p65 isoform is also expressed in a third of different pathological conditions, like benign and inflamed tissue. However, since 83.4% of metastatic head and neck cancer tissues show a strikingly high abundance of BTK-p65, this specific isoform seems to be linked to metastatic processes. In line with this assumption, we previously detected significantly reduced transmigration capacity of the analyzed HNSCC cells upon inhibition of BTK [31], suggesting an influence of oncogenic BTK on migration and invasiveness and, therefore, its involvement in HNSCC metastasis. 

Additionally, in support of this hypothesis, we found that when treated with AVL-292, HNSCC cell lines cultivated under 2D or 3D conditions underwent a profound morphological alteration accompanied by significant changes in the expression of the EMT markers E-Cadherin and Slug upon BTK inhibition. These observations are consistent with previous findings demonstrating that the chemical or genetic abrogation of BTK activity in neuroblastoma cells or oral cell carcinoma cells is associated with increased E-Cadherin and reduced Slug expression [30,36]. A previously conducted meta-analysis revealed that high E-Cadherin expression in HNSCC tumors is associated with better overall survival (OS) and disease-free survival (DFS) [37]. In addition, high expression levels of the transcription factor Slug promoted the EMT and were predictive for DSF [38]. Since treatment with AVL-292 led to an increase in E-Cadherin expression and, at the same time, to a reduction in Slug protein levels, BTK inhibition may represent a powerful tool to interfere with the EMT and metastatic phenotype of HNSCC, thus improving the OS and DFS of head and neck cancer patients. As the EMT is known to be induced by hypoxia in solid tumors, it would also be of great interest to analyze the effect of BTK inhibition on the EMT in HNSCC cells under hypoxic conditions in further studies.

In addition to morphological changes, we found a dramatic decrease in tumor sphere formation under BTK inhibition in all the analyzed cell lines, which was accompanied by a significant reduction in the expression of the cancer stemness marker ALDH1A1. These results support previous findings suggesting BTK to be involved in CSC enrichment in glioblastoma [36] and bladder cancer [39], as well as in OSCC [30]. The aldehyde dehydrogenases (ALDHs) comprise a group of several enzymes that oxidize aldehydes in the metabolism of many living beings. ALDHs are oxidoreductases and ALDH1 is involved in retinal metabolism, which is necessary for cell growth and proliferation [40]. High expression of ALDH1 was associated with the expression of the stemness markers BMI1, OCT4, SOX2, KLF4, and NANOG, which are also characteristically expressed in HNSCC. Furthermore, ALDH-positive cells revealed reduced radio-sensitivity, lower chemotherapy-responsiveness and higher tumorigenic potential in many tumor entities, including head and neck cancer [40,41,42,43]. Thus, the use of BTK inhibitors as a therapeutic option might reduce ALDH-positive CSC and, at the same time, enhance the radio- as well chemo-sensitivity of HNSCC [30]. 

EMT as well as CSC enrichment are processes influenced by chemokines and cytokines produced by cells of the TME [8,43]. This study revealed that HNSCC cells themselves are able to secrete pro-inflammatory cytokines, like IL-6, IFNγ, and TNFα. Previously, Karavyraki and Porter reported the secretion of IL-6, IL-11, and IL-8 by a tongue-derived SCC cell line [44]. Moreover, the authors showed that the survival of tumor cells detached from the extracellular matrix tumor cells, which would normally undergo apoptosis (*anoikis)*, is IL-6-dependent and acts in an autocrine fashion. Antibodies against IL-6 or its receptor were sufficient to revert this resistance to *anoikis*, a marker of EMT [44]. In line with this observation, IL-6 dependent EMT gene regulation was reported in pancreatic cancer cells [45] as well as in HNSCC [46]. In addition, in osteosarcoma, a function of IL-6 in the acquisition and maintenance of stemness characteristics was described [47]. Also, TNFα is able to induce EMT in several cancer entities, including HNSCC [8,48]. Besides the fact that the transcription of most pro-inflammatory cytokines depends on the activity of NF-κB, in HNSCC, TNFα activates the NF-κB signaling pathway, stabilizing Slug and thereby inducing the process of EMT [35]. Together, these data indicate that HNSCC cells secrete pro-inflammatory cytokines involved in the processes of EMT and CSC enrichment, processes which, in turn, depend on the activity of NF-κB. Recently, a function of IFNγ in the EMT has been reported. Analyzing breast cancer, Beziaud et al. found that IFNγ converts non-CSCs into CSCs and enhances their chemo- and radiotherapy resistance [49]. In addition, IFNγ also induces programmed cell death ligand 1 (PD-L1) expression on cancer cells [50], which promotes immunosurveillance and is associated with the poor prognosis of HPV-negative HNSCC patients [51]. Since IL-6, TNFα, and IFNγ are secreted by HNSCC cells, it is likely that HNSCC cells regulate their stemness and EMT characteristics in an autocrine manner independently of cytokines secreted by other cells of the TME.

Moreover, interfering with BTK activity by applying the BTK inhibitor AVL-292 led to a reduction in NF-κB expression and dampened pro-inflammatory cytokine secretion, accompanied by an abrogated the EMT and CSC enrichment, in HNSCC cells cultured under 3D conditions in our study. These findings suggest that BTK directly affects NF-κB-dependent signaling and cytokine secretion, which, in turn influences the EMT and CSC enrichment in HNSCC independently of additional cytokines produced by other cells present in the TME. In B cells, BTK has already been shown to be required for the B cell receptor (BCR)-induced activation of NF-κB [52]. Our findings described here suggest that BTK is also involved in the regulation of NF-κB activity in solid tumors. NF-κB is known to promote the EMT and metastasis in several solid tumor entities, as the expression of various EMT molecules depends on NF-κB [53]. NF-κB is also a repressor of E-Cadherin [54]. Therefore, the here-described increase in E-Cadherin expression upon BTK inhibition might also be a consequence of reduced NF-κB activation. NF-κB is also involved in CSC biology. NF-κB is constitutively activated in CSCs of various tumor entities, promoting tumorigenesis and metastasis [53]. As our data reveal a reduction in CSC enrichment after BTK inhibition, targeting the NF-κB pathway with BTK inhibitors could be a promising tool to inhibit the proliferation and metastasis of CSCs. 

Therefore, our findings highlight the potential of BTK inhibition as a new treatment option for HNSCC and other solid tumor entities. 

## 4. Materials and Methods

### 4.1. Immunohistochemical Staining of Tissue Microarrays and Scoring of BTK Expression

The Head and Neck Tumor Tissue Array was obtained from Biochain (Newark, CA, USA; Catalog No. Z7020051) and contained 48 cases of inflammatory, benign, and malignant tumor tissues of the neck, oronasopharynx, larynx, and salivary glands in duplicate. TMAs were probed with the anti-p65BTK BN30 polyclonal antibody according to standard IHC procedures. The BN30 polyclonal antibody production and characterization was described previously [23]. Slides were also counterstained with Hematoxylin and Eosin to perform pathological evaluation. Samples were blindly analyzed and grouped into negative, low, medium, and high-positive samples. By using an immunohistochemistry Image Analysis Toolbox (Jie Shu, Guoping Qiu, Mohammad Ilyas; the University of Nottingham, UK) for ImageJ (version 1.50i, NIH, Bethesda, MD, USA), different staining intensities were acquired and used as a model to quantify and grade the samples according to increasing intensity.

### 4.2. Cell Lines and Culture

We utilized three well-characterized HNSCC cell lines: UDSCC1, UDSCC5, and UDSCC6 for our experiments. Cells were cultured in both 2D and 3D models. For 2D culture, cells were seeded in Gibco DMEM medium (ThermoFisher Scientific, Waltham, MA, USA) supplemented with 10% FBS (Bio&SELL GmbH, Feucht, Germany), 1% cell shield (Minerva Biolabs, Berlin, Germany), and 1% NEAA (100×, ThermoFisher Scientific, Waltham, MA, USA) in regular culture flasks. For 3D culturing, the floating sphere formation approach using Ultra-Low Attachment (ULA) surface plates (Corning, New York, NY, USA) was applied: cells were resuspended in medium containing DMEM/F12 (Sigma-Aldrich, St. Louis, MO, USA) with B27 Supplement (50×, Sigma-Aldrich), supplemented with 0.4% BSA (Sigma Aldrich), 20 ng/mL EGF (Sigma Aldrich), 10 ng/mL bFGF (Thermo Scientific), and 5 µg/mL human Insulin (Roche, Basel, Switzerland) at a density of 20,000 cells per well on 6-well ULA surface plates and incubated for 72 h at 36 °C.

### 4.3. Drug

AVL-292, purchased from MedChem (Monmouth Junction, NJ, USA), was stored at 4 °C in the dark. Stock solutions at 20 μM concentration were freshly prepared immediately before each use by dissolving in DMSO (Sigma-Aldrich).

### 4.4. Sphere Formation Assay

For the sphere formation assay, cells were seeded at a density of 20,000 cells per milliliter in each well of 6-well ULA surface plates. The cells were cultured in DMEM/F12 medium with supplements (see above). Drugs, if required, were added at this time. After 72 h, the spheres were measured and counted.

### 4.5. Flow Cytometry

Flow cytometry was performed to characterize cancer stem-like cells (CSCs). ALDH1A1, a well-known CSC marker, was detected using the ALDEFLUOR Kit (STEMCELL Technologies, Vancouver, Canada) according to the manufacturer’s instructions. After cell harvesting, the cells were counted, stained, incubated for 40 min, and then washed. The GALLIOS Flow Cytometer (Beckman Coulter, Brea, CA, USA) was used for analysis, and the acquired data were analyzed using Kaluza Analysis software, version 2.1 (Beckman Coulter, Brea, CA, USA).

### 4.6. Western Blotting

Cells were collected after 72 h of treatment with the inhibitor or with DMSO as a control. Cell lysis (with RIPA) and protein quantification (Pierce BCA Protein Assay Kit, Thermo Scientific) were performed. Equal amounts of protein were loaded onto 12% SDS–PAGE gels. Gel electrophoresis and the subsequent Western blotting procedure were performed according to standard protocols. The Western blotting results were quantified using Image Lab software (version. 6.0.1., BIO-RAD, Hercules, CA, USA). The following antibodies were used: E-Cadherin (24E10), rabbit monoclonal antibody (Cell Signaling Technology, Danvers, MA, USA); Slug (C19G7), rabbit monoclonal antibody (Cell Signaling); ß-actin (ab8227), rabbit polyclonal antibody (abcam, Cambridge, UK); and NF-κB p65 (C-20), rabbit polyclonal antibody (Santa Cruz Biotechnology, Dallas, TX, USA).

### 4.7. RNA Sequencing (RNA-Seq) and Analysis

RNA-Seq was employed to profile and quantify the entire complement of RNA molecules present in the HNSCC samples. Total RNA was extracted from the spheroids generated from UDSCC1, -5, and -6 cells on ULA surface plates. After their quality was checked using Agilent Tape Station and their quantity measured by the Qubit Fluorometer (Invitrogen, Waltham, MA, USA), 200 ng of each RNA was used to generate sequencing libraries using the Illumina RNA-Seq Kit V2. Sequencing libraries were quality checked on D1000 screen tapes (Agilent Technologies, Santa Clara, CA, USA), and samples were sequenced using a NextSeq550 (Illumina, San Diego, CA, USA) and 2 NextSeq 500/550 High Output Kits v2.5 (75 Cycles) at the Genomics Core Facility (Ulm University). High-quality reads were mapped to the human genome (hg38) using STAR (2.0.9). This was followed by the removal of multimapping reads and the conversion to gene-specific read counts for annotated genes using featureCounts (2.0.0). Data analysis was performed in R (4.0.2) using the IDE Rstudio (1.3.1056). Clustering in the heatmaps was performed using hierarchical clustering within the heatmap (1.0.12) package.

### 4.8. ELISA

Cytokine release was detected in supernatants of HNSSC cells cultivated under 3D conditions according to standard protocols. Antibodies against IL-6 (5IL6, ref M620), IFNγ (B133.5, ref M701B), and TNFα (28401, ref MA5-23720) were purchased from Invitrogen. The recombinant human cytokine standards were from Repro Tech. Extinction was analyzed at 405/490 nm on microplate ELISA reader.

### 4.9. Statistical Analysis

For statistical analysis, GraphPad Prism version 8.4.3 was used. The two-tailed *t*-test or Mann–Whitney U-test was applied as indicated. At least five independent experiments were performed. Data are presented as the mean +/− standard deviation. *p* < 0.05 was considered significant (* *p* < 0.05, ** *p* < 0.01, *** *p* < 0.001, **** *p* < 0.0001).

## 5. Conclusions

We describe the expression of the oncogenic BTK isoforms p80 and p65 in head and neck tumor tissue, whereby BTK-p65 is highly abundant in metastatic head and neck tumor cases. Complementarily, we observed changes in cell morphology and in the expression of EMT markers upon BTK inhibition. Treatment with the BTK inhibitor AVL-292 also reduced tumor sphere formation in HNSCC-derived cell lines and the expression of the stemness marker ALDH1A1. In addition, BTK inhibition reduced NF-κB expression as well as NF-κB-dependent cytokine secretion from HNSCC cells. Altogether, BTK inhibition reduced NF-κB activity, pro-inflammatory cytokine release, and impaired the EMT and CSC enrichment in HNSCC. Thus, our data demonstrate a high potential to treat HNSCC patients with BTK inhibitors, either in combination with today’s applied standard-of-care therapies or with novel immunotherapeutic approaches, in order to improve the response to therapy and thus make it more satisfactory.

## Figures and Tables

**Figure 1 ijms-24-13133-f001:**
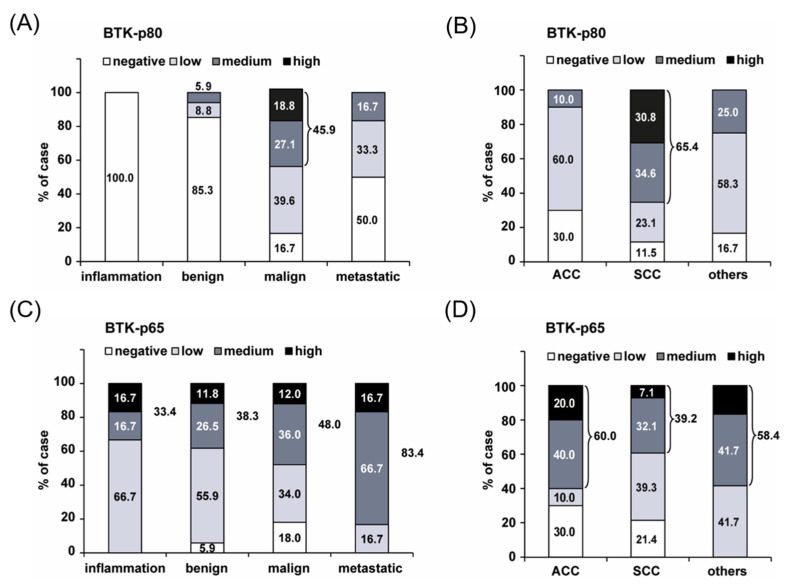
BTK-p80 and BTK-p65 are differentially expressed in tumor samples of head and neck tumor tissue. Tissue microarray (TMA) was stained for BTK-p80 using the BTK 7F12H4 antibody obtained from Santa Cruz, which detects BTK-p80 only [31] (**A**,**B**), or for BTK-p65 using the BTK BN30 antibody [22] (**C**,**D**). Quantification of the immunohistochemical staining is depicted. Percentages of positively stained cases were grouped according to inflamed, benign, malignant, and metastatic tissue. (**B**,**D**) BTK expression was analyzed in different head and neck tumor entities like adenoid cystic carcinoma (ACC) and squamous cell carcinoma (HNSCC). Overall, the TMA contained 48 cases in duplicate: 3 cases of inflamed, 17 of benign, 25 of malignant, and 3 of metastatic tissues.

**Figure 2 ijms-24-13133-f002:**
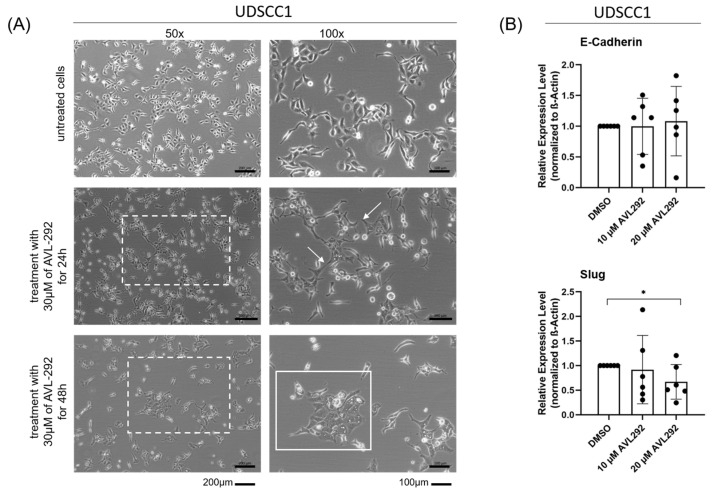
Inhibition of BTK with AVL292 leads to morphological changes in two-dimensional cultures of HNSCC cells. UDSCC1 cells were treated with 10 µM, 20 µM, and 30 µM AVL-292 or 0.1% DMSO as a control over a period of 24 or 48 h. (**A**) Pictures of phase contrast microscopy are shown. Dashed boxes show the section, which is shown in 100× magnification on the right side. Arrows show cell extensions and cell islands are marked by solid boxes. (**B**) Protein lysates of treated cells were analyzed by Western blotting using antibodies against E-Cadherin and Slug. Image Lab (BioRad) was used as an analyzing tool. Relative expression levels were determined by normalizing against beta-actin levels. The graphs summarize the results of five different experiments. * *p* < 0.05.

**Figure 3 ijms-24-13133-f003:**
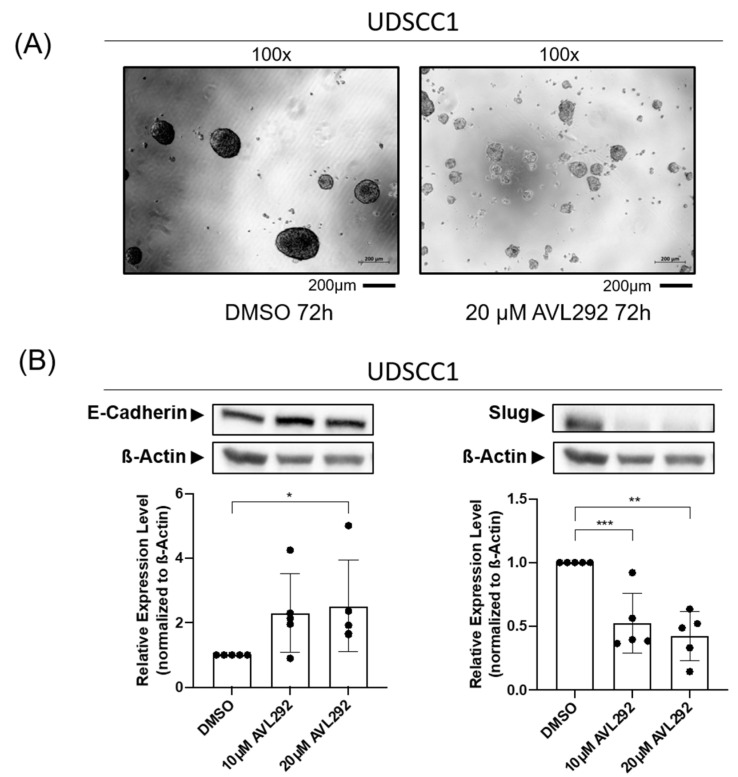
Inhibition of BTK with AVL292 leads to morphological changes in 3D cultures of HNSCC cells. UDSCC1 cells were cultured under 3D conditions on ultra-low attachment surface plates and treated with 10 µM and 20 µM AVL292 or 0.1% DMSO as a control over a period of 72 h. (**A**) Pictures of phase contrast microscopy are shown. (**B**) Protein lysates of treated cells were analyzed by Western blots using antibodies against E-Cadherin and Slug. Image Lab (BioRad) was used as an analyzing tool. Relative expression levels were determined by normalizing against beta-actin levels. The graphs summarize the results of five different experiments. * *p* < 0.05, ** *p* < 0.01, and *** *p* < 0.001.

**Figure 4 ijms-24-13133-f004:**
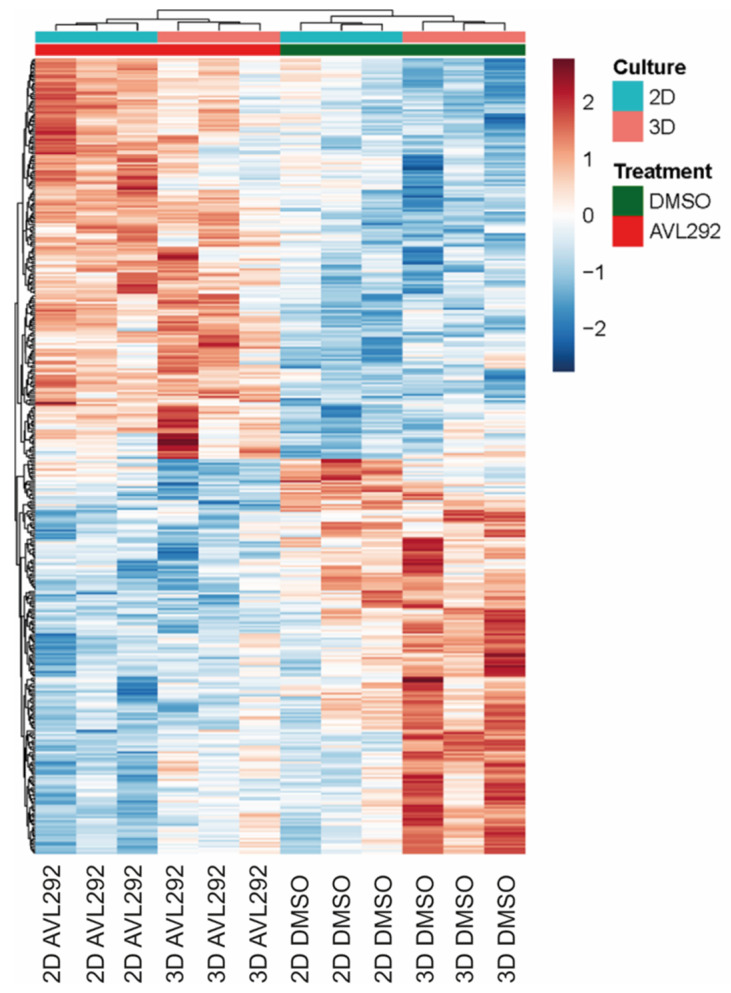
Hierarchical clustering in the heatmap shows differentially expressed genes between HNSCC cells (UDSCC 1, 5, and 6) treated with 20 µM AVL292 or 0.1% DMSO as a control for 72 h. Cells were either cultured under 2D conditions or on ULA surface plates as a three-dimensional culturing system. Data analysis was performed in R (4.0.2) using the IDE Rstudio (1.3.1056).

**Figure 5 ijms-24-13133-f005:**
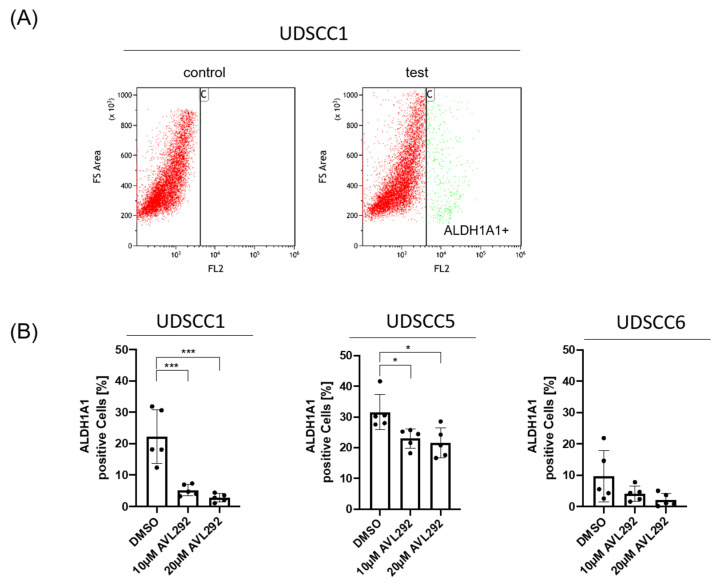
Inhibition of BTK by AVL292 in spheres of HNSCC cell lines leads to changes in the expression of the cancer stem cell marker ALDH1A. HNSCC1, 5, and 6 were cultivated on ultra-low attachment surface plates, treated with 10 µM and 20 µM AVL292 or 0.1% DMSO as a control, and collected after 72 h. (**A**) FACS analyses were performed using a flow cytometer (Gallios; Beckman Coulter) and Kaluza software as an analyzing tool. A characteristic experiment is presented. (**B**) An ALDEFLOUR assay was performed to quantify ALDH1A1-positive cells in spheres of analyzed cell lines as indicated. * *p* < 0.05, and *** *p* < 0.001.

**Figure 6 ijms-24-13133-f006:**
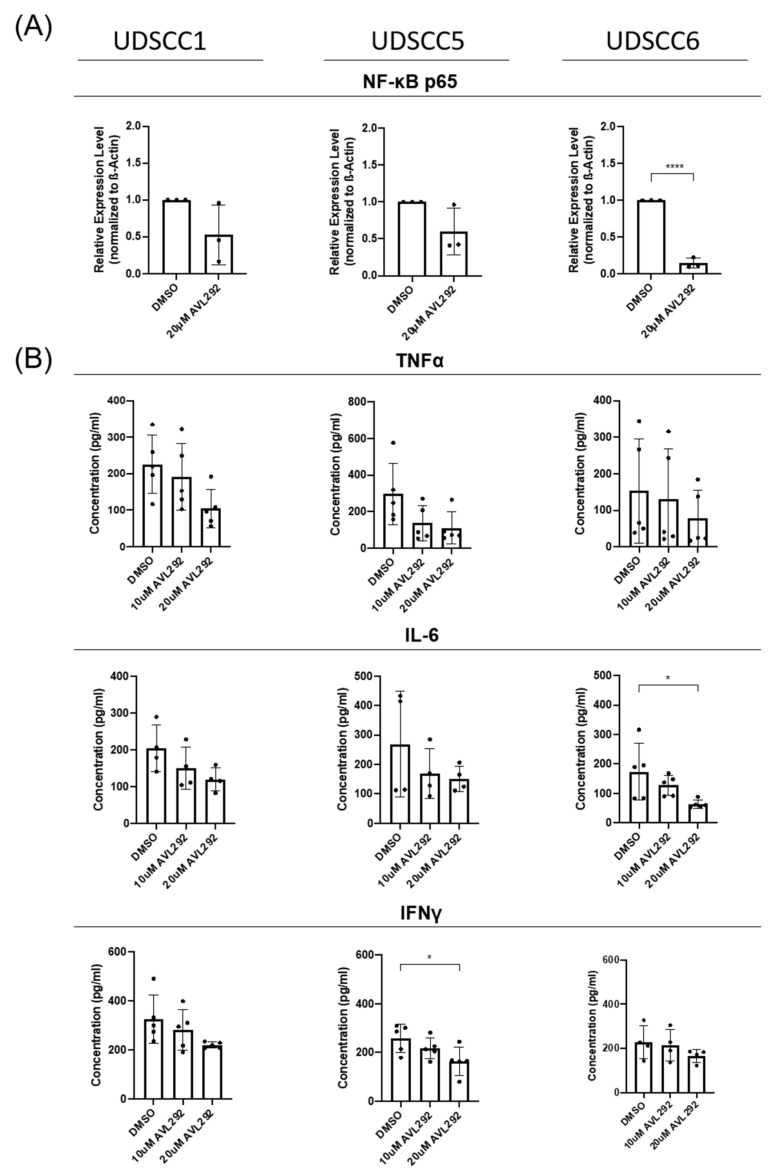
In 3D-cultured HNSCC cells, the NF-κB-pathway and cytokine release are affected by BTK inhibition. UDSCC1, -5, and -6 cells were cultivated on ultra-low attachment surface plates and treated with 10 µM or 20 µM AVL-292 (as indicated) or 0.1% DMSO as a control over a period of 72 h. (**A**) Protein lysates of treated cells were analyzed by Western blotting using an antibody against NF-κB p65. Image Lab (BioRad) was used as an analyzing tool. Relative expression levels were determined by normalizing against actin levels. The graphs summarize the results of 5 five different experiments. (**B**) Supernatants of the spheres were collected and ELISA was performed on these using antibodies against TNFα, IL-6, and IFNγ. * *p* < 0.05, and **** *p* < 0.0001.

## Data Availability

Original data are available upon request.

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
