# Peer review of "Influence of Bruton’s Tyrosine Kinase (BTK) on Epithelial–Mesenchymal Transition (EMT) Processes and Cancer Stem Cell (CSC) Enrichment in Head and Neck Squamous Cell Carcinoma (HNSCC)"

_ijms, 2023, doi:10.3390/ijms241713133_

Round 1
Reviewer 1 Report
The manuscript by Leichtle et al deals with the Bruton's Tyrosine Kinase (BTK) and its importance in the HNSCC, specifically EMT process and CSC enrichment. Overall, the manuscript is well written and the data are clearly presented.
Some concerns:
1) Is the AVL-292 specific to BTK p80? Are both isoforms of BTK important in HNSCC in regard to treatment?
2) EMT is known to be induced by hypoxia in solid tumors. Do authors know the effect of BTK inhibition in hypoxic cells?
3) Authors write that the HNSCC cells were cultured in medium supplemented with 10%FBS. Serum is known to severely affect CSC phenotype. This should be taken into consideration.
Reviewer 2 Report
The study demonstrates the effect of Bruton's Tyrosine Kinase on epithelial-mesenchymal transition and cancer stem cell enrichment in head and neck squamous cell carcinoma. BTK inhibition supressed the NF-kB pathway and cytokine release.
Figure 4 may be revised to indicate the name of the columns with cell cultures.
ELISA and statistical analysis can be described in separate sections rather than in 4.7. RNA Sequencing (RNA-seq) and analysis.
Author Response
Figure 4 may be revised to indicate the name of the columns with cell cultures.
Thank you very much for taking time to review our manuscript. We revised Figure 4, which now includes the name of each column with corresponding cell cultures and treatments.
ELISA and statistical analysis can be described in separate sections rather than in 4.7. RNA Sequencing (RNA-seq) and analysis.
Thank you very much for this hint. The ELISA and statistical analysis part were grouped to the 4.7 RNA-Seq section by mistake and are now described in separate sections 4.8. and 4.9 (now 5.8 and 5.9 as the Material & Methods section was shifted due to the insertion of the Conclusions section).
Reviewer 3 Report
I greatly appreciate the opportunity to review the manuscript titled "Influence of BTK on EMT processes and CSC enrichment in HNSCC." I would like to extend my congratulations to the authors, as I firmly believe that this work constitutes a valuable addition to the current state of knowledge regarding Bruton's Tyrosine Kinase. Below, I will describe the minor issues I have observed.
1. I believe that the extensive use of abbreviations in the title of the paper makes it difficult to decipher at first glance. I urge the authors to consider using full phrases.
2. The link between BTK and NF-κB is insufficiently described. BTK is involved in NF-κB signaling in a way that is significantly more complex than depicted. I believe that although there is no need to describe it extensively, it should be mentioned nevertheless.
3. I would appreciate the addition of a summary of the discoveries, either at the very end of the Discussion or as a separate section in the Conclusions.
Author Response
- I believe that the extensive use of abbreviations in the title of the paper makes it difficult to decipher at first glance. I urge the authors to consider using full phrases.
Thank you very much for taking time to review our manuscript. We agree that using abbreviations already in the title makes it difficult for the reader. Therefore, we are using full phrases now in the title of the revised version of our manuscript.
- The link between BTK and NF-κB is insufficiently described. BTK is involved in NF-κB signaling in a way that is significantly more complex than depicted. I believe that although there is no need to describe it extensively, it should be mentioned nevertheless.
Yes, you are absolutely correct that the interplay of BTK and NF-kB was insufficiently described and is certainly more complex than described. Therefore, we now added a paragraph explaining the link between BTK and NF-kB in more detail to the Discussion section (lines 337-354).
- I would appreciate the addition of a summary of the discoveries, either at the very end of the Discussion or as a separate section in the Conclusions.
Yes, we agree and included a separate Conclusions section after the Discussion section where we now summarize our findings.